# A Novel Method to Estimate the Full Knee Joint Kinematics Using Low Cost IMU Sensors for Easy to Implement Low Cost Diagnostics

**DOI:** 10.3390/s20061683

**Published:** 2020-03-18

**Authors:** Mark Versteyhe, Henri De Vroey, Frederik Debrouwere, Hans Hallez, Kurt Claeys

**Affiliations:** 1Department of Mechanical Engineering, Division RAM, M-Group, KU Leuven Campus Bruges, 8200 Bruges, Belgium; mark.versteyhe@kuleuven.be; 2Department of Rehabilitation Sciences, We-Lab for HTM, KU Leuven Campus Bruges, 8200 Bruges, Belgium; henri.devroey@kuleuven.be (H.D.V.); kurt.claeys@kuleuven.be (K.C.); 3Department of Computer Sciences, imec.Distrinet, M-Group, KU Leuven Campus Bruges, 8200 Bruges, Belgium; hans.hallez@kuleuven.be

**Keywords:** knee joint, kinematics, IMU, joint angle

## Abstract

Traditional motion capture systems are the current standard in the assessment of knee joint kinematics. These systems are, however, very costly, complex to handle, and, in some conditions, fail to estimate the varus/valgus and internal/external rotation accurately due to the camera setup. This paper presents a novel and comprehensive method to infer the full relative motion of the knee joint, including the flexion/extension, varus/valgus, and internal/external rotation, using only low cost inertial measurement units (IMU) connected to the upper and lower leg. Furthermore, sensors can be placed arbitrarily and only require a short calibration, making it an easy-to-use and portable clinical analysis tool. The presented method yields both adequate results and displays the uncertainty band on those results to the user. The proposed method is based on an fixed interval smoother relying on a simple dynamic model of the legs and judicially chosen constraints to estimate the rigid body motion of the leg segments in a world reference frame. In this pilot study, benchmarking of the method on a calibrated robotic manipulator, serving as leg analogue, and comparison with camera-based techniques confirm the method’s accurateness as an easy-to-implement, low-cost clinical tool.

## 1. Introduction

In order to diagnose patients, plan surgical procedures and adapt rehabilitation programs, a dedicated movement analysis laboratory equipped with infrared cameras, combined with reflective markers attached on the body, has typically been used as a gold standard for the assessment of human motion (e.g., gait analysis or functional movement analysis) [1]. However, this motion analysis of subjects within a laboratory environment has been related to many limitations (e.g., complex handling of equipment, spatial constrains, and the variability in marker placement affecting the final kinematical outcome). Therefore, new devices have been developed to address these shortcomings. Inertial measurement units (IMUs) are light weight sensors, including a magnetometer, gyroscope, and accelerometer, that measure and report a body’s specific force, angular rate, and sometimes the orientation of the body. IMUs were originally used into aircraft industry in order to calculate altitude and position of the manned or unmanned aircraft or satellite. Recently, the capacities of IMUs to detect positions and accelerations was introduced into biomedical or movement sciences aiming to perform a precise motion analysis of the body or parts of the human body where they are attached. These possibilities open some windows for the field of the movement of rehabilitation sciences as IMUs are not dependent on a specific movement analysis laboratory. They enable researchers to perform motion analysis in an outside lab situation, such as in a hospital or on a sports field [2].

For instance, De Vroey et al. [3] succeeded to validate the use of IMUs with an optical motion capture system for the calculation of temporal aspects of gait in a knee arthroplasty population. Despite the excellent possibilities for the use of IMUs in motion analysis, some crucial concerns and methodological problems need to be tackled especially to use them for estimation of joint kinematics. As stated in a review by Weygers et al. [4], IMUs can be used to evaluate joint kinematics of the lower limb, but combined efforts of engineers and movement experts are still necessary to develop scripts to calculate joint angles based on gyroscope and accelerometer data, which is still a big challenge. Moreover, developed scripts need to be validated with a golden standard for motion analysis before implementation in clinical setting (e.g., patients, sportsmen, etc.).

Focusing on the knee joint, many authors used IMU sensors for the analysis of the main flexion/extension angle. An extensive and excellent literature review on these methods has been provided by Hallez et al. and Seel et al. [5,6]. One method by [6] is of particular interest as it splits the estimation of the flexion angle in two steps. In this paper, a similar approach to the method used by [6] is proposed because it generalized and extended such that it also yields precise kinematical estimations in both the frontal (varus/valgus) and transverse plane (internal/external rotation). One of the challenges into the use of IMUs for motion analysis is the extensive calibration, which should in an ideal way be shortened and not be dependent of the sensor placement.

The main aim of this study was to develop a valid method to use IMUs in the evaluation of knee joint kinematics in the sagittal (flexion-extension), the frontal (varus-valgus), and the transversal (internal-external rotation) plane. The main specific aim was to achieve an easy-to-use and portable clinical analysis tool.

## 2. Description of the Method

Current state-of-the-art IMU based methods require an extensive calibration procedure, which in turn can be quite difficult to perform and tiring for some patient groups. The proposed method tackles this problem by a quick calibration of the sensors and the advantage to arbitrarily place the sensors on the leg segments (respecting some guidelines). This makes the proposed method very user friendly to clinicians, who are typically not familiarized with the calibration of complex motion capturing systems.

In the proposed method, IMU (Inertial and Gyroscopic Unit) sensors are placed on the upper and lower leg by a physiotherapist. The acceleration and gyroscopic sensor readouts are recorded for analysis while the patient executes leg motions as required for clinical or scientific purpose. As many other authors state, the information of the magnetic sensor is of no use due to local field changes in concrete reinforced buildings and the presence of ferromagnetic components. The sensors are arbitrarily placed on the limbs; hence, it does not suffice to subtract the gyroscope angles to obtain the varus/valgus, internal/external rotation and flexion/extension angles. Furthermore, integrating the raw IMU data, as often stated to be straightforward, to obtain the IMU frame angles is prone to integration error and drift. The proposed algorithm uses the raw IMU data (i) in an optimal way, (ii) based on a kinematic leg model, (iii) by accepting that the IMU sensors are arbitrarily located on the limbs, and (iv) by using both angular velocity data and accelerometer data of both IMU sensors.

The proposed method utilizes a two-step analysis, after which the results are presented in a reference frame of choice.

Firstly, the knee joint position and direction with respect to the sensor frames connected to the upper and lower leg are estimated. Minor improvements to existing methods are made in order to render the estimation more robust. The upper and lower leg are considered to be rigid body objects while the knee joint is considered to be an elliptical joint where rotation happens preferably around the mediolateral (knee) axis but also allows slight rotation around the anteroposterior axis.

Secondly, a simple time model is built which describes the rotational motion of the legs as rigid bodies, connected to the sensor frames. Furthermore, the model includes physical constraints (i) between those rigid bodies and (ii) between the rigid bodies and the world reference. A Rauch Tung Striebel filter [7] (An RTS filter is a fixed interval smoother that has a recursive implementation) is used to yield the motion of the upper and lower leg in world reference coordinates. The constraints are judicially chosen such that they eliminate drift under a few mild assumptions about the motion of the patient. The RTS filter shows superior performance (see further) over earlier applied Kalman filter based methods (by the authors of the presented work), as it is able to propagate information of future knowledge to the past and reduces the uncertainty on the results significantly. The RTS also shows superior performance in estimating the minor angles (varus/valgus and internal/external rotation), accurate enough to be used for diagnostic purpose (see further).

### 2.1. Proposition

Quaternion math is used to describe rotation of the body and sensor frames under interest as they allow for a simple compact representation. The relation of the derivation of quaternions to instantaneous rotation around the axis and the algebra involved in converting to and from quaternion frames is excellently covered by [8,9] and is not repeated here for brevity. The orientation of the frames follows the convention of [6] and shown in Figure 1. In this paper, we only treat the right leg—the necessary adaption (as the reference frame is different) to treat the left leg is trivial.

As the internal sensor frame of the accelerometer and gyroscopes do not perfectly align with the sensor enclosure, some sensors offer the feature to zero the sensor: a procedure that provides the results in a frame where the first axis aligns with the gravity axis at the time of pressing a button. The sensor can now be “zero-ed” in two possible ways. One can zero the sensor after connecting the sensor to the patient OR one can zero the sensor frame when aligning it to a perfectly vertical reference line. As we will see, we prefer the latter as it helps to estimate the initial conditions, but the first approach can be chosen, as well, and only requires rotating the results in the proper frame during post-processing. In the following text, we assume that the IMU’s sensor frame is aligned with (i) the vertical gravity axis and, (ii) secondly, more or less in line with the patients general direction of motion during zeroing. The physiotherapist does this by fixing the sensor to an artefact that is set level vertically and more or less in the walking direction and by pressing a zero button. After that, the physiotherapist rigidly secures the IMU to the lateral aspect of the thigh, along a reference line between the greater trochanter and lateral epicondyle (representing the actual orientation of the femoral bone). In turn, the IMU of the lower leg is fixed to the lateral aspect of the shank, along a reference line between the head of the fibula and lateral malleolus. The patient then executes the dynamic motion task that is to be analyzed on a more or less straight line. The method does not allow for the patient to move along a random or arbitrary curved path. That said, this constraint can be relaxed if specific motion for diagnostic purpose, e.g., on a circle, is absolutely needed. In that case, the therapist could indicate to the program along which path the patient moves (e.g., enter the curvature of the line) at which time instant. It is not implemented by the authors as it does not add scientific value, but—if important—it could be done.

The method is scripted in MATLAB mathsizesmall and is publicly available with measurement data, and it is free to download and use at the website of the research group GROUP mathsizesmall M [10]. The authors invite you to comment and add or adapt the code to your specific need(s).

Throughout the paper following notations are used. Frame {A} determines the rotation of the sensor frame {A} with respect to the world reference frame {w}. $q_{w}^{A}$ then stands for the quaternion rotating the world reference frame into the reference frame {A}. Frame {A} is rigidly connected to the upper leg, while frame {B} is connected to the lower leg. This setup is depicted in Figure 1.

### 2.2. Conditioning of the Measurements

Measurements are obtained from a wireless IMU sensor (MTw Xsens), with a sampling rate of 120 Hz. The IMU sensor gyroscope and accelerometer are pre-calibrated to remove bias. The sensor is firmly attached to the leg—a quick dynamic analysis shows that there is no evidence that the sensor moves with respect to the leg. High frequent noise is, however, substantial and will (i) prohibit the numerical determination of derivatives and (ii) disturb the estimation process of the joint flexion/extension axis of the body members. Therefore, the measurements are first filtered with a third order Butterworth zero phase filter set at 30% of the sampling frequency, filtering out noise but preserving all relevant details. Next, the derivatives of the angular velocity are calculated using Simpsons’s approximation
$${\dot{\omega }}_{k}\approx \frac{1}{12}\left(\omega_{k-2}-8\omega_{k-1}+8\omega_{k+1}-\omega_{k+2}\right),$$
where $\dot{\omega }$ is the derivative of the momentary angular velocity ω around each of the three axis of the sensor reference frame. This method is used for all six angular components.

Measured variables by the IMU sensors are the angular velocity ω and linear accelerations *a* of both frames {A} and {B}.

Note on the mounting of the sensors on the test subject. The single-band mechanical connection of the sensor to the leg of the patient are specifically developed for these applications. The band is not fully rigid but semi-elastic. This ensures the sensor keeping its optimal and initial position during the movement, despite the change in volume of the underlying contracting muscles during movement. The elasticity also allows muscles to contract and relax, without fixation of blood flow or causing pain. The inside (contact with the body) is developed in a silicon material, which ensures no glide nor translational displacement of the sensor during movement, which is very important for reliable motion analysis. The outside consists of velcro, enabling very good fixation of the sensor on the band during movement.

### 2.3. Determination of Orientation of the Average Knee Axis in Both A and B Reference Frames

The orientation of the joint axis is derived in much the same way as described in detail by [6] but extended for faster convergence and robustness. In short, this method determines the knee axis by minimizing a cost function which represents (i) the difference between the remaining rotational speed perpendicular to the sought common joint bending axis and (ii) (as an extension to the algorithm by [6]) the mismatch of acceleration along the joint axis and perpendicular to that joint axis at the location of the joint axis. This cost function is depending on four degrees of freedom (two for the orientation of the common knee axis in each reference frame {A} and {B}), which are to be obtained from minimization.

The first element in the cost function is described by
$${\cos t}_{1}=∥\omega_{A}\times j_{1}^{A}∥-∥\omega_{B}\times j_{2}^{B}∥,$$
where $j_{1}^{A}$ describes the sought axis in the {A} reference frame, and $j_{2}^{B}$ the sought (same) axis in the {B} reference frame. If the upper and lower leg rotate mainly around, respectively, $j_{1}^{A}$ (or $j_{2}^{B}$), the magnitude of the remaining rotational component should be similar in frame {A} and frame {B} since both axis in the one frame and in the other frame coincide. The orientation of the two remaining rotational components with respect to frame {A} and frame {B}, however, cannot be compared as the orientation for the moment is unknown. The second element in the cost function is based on the fact that the acceleration in the joint point expressed in both reference frames A and B should be the same:$${\cos t}_{2}=\left|a_{1⊥j}-a_{2⊥j}\right|+\left|a_{1∥j}-a_{2∥j}\right|,$$
where
$$\begin{array}{c} a_{1⊥j}=j_{1}^{A}\cdot a_{j_{1}}^{A},\;\;\;\;a_{1∥j}=∥j_{1}^{A}\times a_{j_{1}}^{A}∥,\\ a_{2⊥j}=j_{2}^{B}\cdot a_{j_{2}}^{B},\;\;\;\;a_{2∥j}=∥j_{2}^{B}\times a_{j_{2}}^{B}∥. \end{array}$$

Hereby,
$$\begin{array}{ccc} a_{j_{1}}^{A} & = & a_{A}+\omega_{A}\times \left(\omega_{A}\times p_{1}^{A}\right)+{\dot{\omega }}_{A}\times p_{1}^{A},\\ a_{j_{2}}^{B} & = & a_{B}+\omega_{B}\times \left(\omega_{B}\times p_{2}^{B}\right)+{\dot{\omega }}_{B}\times p_{2}^{B} \end{array}$$
is a good approximation of the acceleration at the joint axis. Of the three components of that acceleration vector we only know one direction in both reference frames: the direction of the axis. The (two other) acceleration components of the acceleration vector perpendicular on that (joint) axis can only be compared on magnitude as the orientation of {A} with respect to {B} we do not know yet. Hence, the absolute value of both cost components are added and minimized. The result is $j_{1}^{A}$ and $j_{2}^{B}$ describing the main rotating axis in each coordinate frame of the knee joint.

Note 1. This cost function requires knowledge of $p_{1}^{A}$ and $p_{2}^{B}$, which are the translation location of the joint axis with respect to the sensor frame origin. The algorithm starts here with an estimate of that position, then determines the orientation of the axis, as we will see in the following section. Next, it determines the exact position using the orientation of the axis. Finally, the algorithm then will reiterate finding the orientation of the axis.

Note 2. For additional robustness, the costs are weighed (pre-multiplied) with the average rotational speed as a function of time. In this way, sections in time where there is no motion (e.g., when the patient waits to start), hence not containing a lot of meaningful data, are not considered by the minimization algorithm.

### 2.4. Determine the Relative Position of the Average Knee Axis in Both A and B Reference Frames

Finding the position of the closest point on the joint axis with respect to the sensor origin is again formulated as a minimization problem. The cost function which represents this is the difference in magnitude of the acceleration of the joint point:$${\cos t}_{3}=∥a_{j1}^{A}∥-∥a_{j2}^{B}∥.$$

The sought position of that point (in both reference frames) influences the result to a larger extent when there is relative motion and/or rotational acceleration of one of the limbs, i.e., the sensor acceleration deviates more from the gravity. Therefore, this cost function is weighted with the deviation of gravity so that more attention is given to measurement data containing meaningful information about that position. As stated earlier, having a better estimate of the position the orientation of the joint axis can be iterated.

### 2.5. Determining Simple Time Model and Output Models

As mentioned above, it is targeted to use an RTS filter for angle estimation. As this filter requires a discrete time state space model of the leg movement and some output models, they are derived in this section. In this paper, in contrast to classical usage of RTS filters, the system measurements are not only used in a measurement update but are incorporated in the system model step. Furthermore, the output models of the system, used in the ‘measurement’ update step, are actually formulated as constraints on the states of the system.

#### 2.5.1. Model

The eight states are the two times four elements of the quaternions $q_{w}^{A}$ and $q_{w}^{B}$ that describe the relative attitude of the frames with respect to the world. The derivation of the discrete time model (including modeling error ϵ)
(1)$$\begin{array}{ccc} q_{w_{k+1}}^{A} & = & \frac{q_{w_{k}}^{A}+\frac{1}{2}\left(q_{w_{k}}^{A}⊗\Omega_{k+1}^{A}\right)dt}{∥q_{w_{k}}^{A}+\frac{1}{2}\left(q_{w_{k}}^{A}⊗\Omega_{k+1}^{A}\right)dt∥}+\epsilon ,\\ & = & f(q_{w_{k}}^{A},\Omega_{k+1}^{A})+\epsilon \end{array}$$
is straightforward. Here, *q* are states *X* and Ω are system inputs *U*. The prediction accuracy was improved by implementing more sophisticated approximations for the model (Equation 1) moving from trapezoid integration rules to Simpson approximations. It can be concluded that the remaining error on the results presented to the user is only for a small amount related to the integration rule; therefore, the most simple is chosen for implementation (represented by Equation (Equation 1)). In this model,
$$\Omega =\left[\begin{array}{cccc} 0 & \omega_{x} & \omega_{y} & \omega_{z} \end{array}\right],$$
is the real instantaneous rotation vector in quaternion notation and ⊗ indicates the quaternion product. Note that we introduce here a measurement signal into the model directly. We can show, however, on the one dimensional case, that the model error is a member of a Normal distribution.
$$q=q+\omega_{\mathrm{model}}dt-\left(\underset{\in \;N(0,\rho )}{\underset{︸}{\omega_{\mathrm{model}}dt-\omega_{\mathrm{real}}dt}}\right).$$

The initial uncertainty is set differently for the four elements of the quaternion. The first element is the cosine of the half angle, while the further three elements also contain the direction vector. As the uncertainty on the states quickly narrow from the initial uncertainty, the initial value does not play a large role, and we can set an overestimate to the initial uncertainty. Constraints on the results will be formulated as extra outputs of the model that need to be zero.

#### 2.5.2. Output Model Constraints

As the motion must satisfy constraints, these will be used to update the model estimates accordingly. The motion constraints in this case are: (i) an estimate of the acceleration measured by the sensors should be equal to the rotational position $q_{w}^{A}$ and $q_{w}^{B}$ of the upper and lower leg, respectively; (ii) the difference of acceleration of the mutual point on the joint axis as described by measurements related to upper leg and lower leg, both noted in world coordinates that should be near zero; (iii) the difference of the orientation of the joint axis as described by measurements from the upper and lower limb again in world coordinates, which should be small; and (iv) the Euler angle around the vertical world axis, which should more or less be constant (and arbitrary chosen).

As typically done in RTS and KF, model updates are performed by comparing output model equations (based on states) with measurements. The motion constrains will therefore be formulated as output equations where the measurement is artificial in case of a true constraints, i.e., for (ii)–(iv).

The first output of the model is a block of six individual outputs (three for each frame) and is pending the attitude of the two frames and needs to be compared with the measurement of the accelerometers in the reference frame of the respective sensors
$$z_{1}=\left[\begin{array}{c} q_{w}^{A}⊗q_{g}\\ q_{w}^{B}⊗q_{g} \end{array}\right]=g_{1}\left(\underset{states}{\underset{︸}{q_{w}^{A},q_{w}^{B}}}\right),\;\;\mathrm{where}\;z_{1}=\mathrm{measured}\left[\begin{array}{c} a_{A}\\ a_{b} \end{array}\right].$$
here, $q_{g}=\left[\begin{array}{cccc} 0 & 9.81 & 0 & 0 \end{array}\right]$. The uncertainty on the measurement is set low when there is no rotation and very high when one of the limbs moves. When there is motion, other acceleration components next to gravity affect the accelerometer reading (as is detailed out by the next output group), and the information cannot be used. This output of the model corrects for slow drift of the quaternions, with the exception of rotational drift around the vertical world axis, which the accelerometers cannot detect.

The second output is the difference between the two acceleration vectors at the joint point on the joint axis rotated into the world reference frame. It is a group of three numbers. Ideally, the difference is perfectly zero. The error can correct the relative rotation between both quaternions and reduces the error on both. The uncertainty level in the covariance matrix related to this output is experimental determined and pends (i) errors on the identification of the joint average axis and (ii) sensor inaccuracies. We observe that comparing the normalized acceleration values provides better results than comparing the absolute components directly. The authors assume that the small calibration errors on the individual readouts of the three acceleration sensor signals are the underlying reason.
$$\begin{array}{ccc} z_{2} & = & \left[\begin{array}{c} q_{w}^{A}⊗a_{j_{1}}^{A}-q_{w}^{B}⊗a_{j_{2}}^{B} \end{array}\right]\\ & = & g_{2}\left(\underset{states}{\underset{︸}{q_{w}^{A},q_{w}^{B}}},\underset{parameters}{\underset{︸}{p_{1}^{A},p_{1}^{B}}},\underset{measured}{\underset{︸}{\omega_{A},\omega_{B},a_{A},a_{b}}}\right),\\ & = & \mathrm{artificially}\;\mathrm{set}\;\mathrm{equal}\;\mathrm{to}\;0. \end{array}$$

The third output is the difference of the orientation of the joint axis as described by the upper limb and the lower limb in the world reference frame. This output has three components and should be small (with average zero) as the main flexion motion happens around one single axis. The secondary motions (varus/valgus and internal/external rotation) are the main reason why the two axis $j_{1}^{A}$ and $j_{2}^{B}$ in world coordinates differ. The uncertainty on this artificial constraint is set such that, when the flexion/extension angle is near zero (or any arbitrary identifiable flexion/extension orientation between the upper and lower leg), it is small. In contrast, the uncertainty while swinging the leg is set very large. In this way, fast (in step) variations of the two rotational axis with respect to another by varus/valgus or internal/external rotation are possible, while each time the leg is vertical (or in that other selected arbitrary position), the relative rotation between the two limbs around the world vertical axis is “reset” so that the two joint axis described in their respective frames do not drift apart over time. Taking this into account lowers slightly the overall error on the states but, more importantly, ensures that the rotational component of the upper and lower leg around the world vertical axis does not drift apart. Note that the motion of the patient does not necessarily need to be on a straight line here. This constraint prohibits drift of the two members around the vertical component with respect to another, regardless of whether or not the motion is executed on a straight line.
$$\begin{array}{ccc} z_{3} & = & \left[\begin{array}{c} q_{w}^{A}⊗j_{1A}-q_{w}^{B}⊗j_{1B} \end{array}\right]\\ & = & g_{3}\left(\underset{states}{\underset{︸}{q_{w}^{A},q_{w}^{B}}},\underset{parameters}{\underset{︸}{j_{1A},j_{1B}}}\right)\\ & = & \mathrm{artificially}\;\mathrm{set}\;\mathrm{equal}\;\mathrm{to}\;0. \end{array}$$

The fourth output is the Euler angle or the rotation of the upper limb around the vertical world axis (gravity). The fourth output is compared to an arbitrary but fixed value (here chosen to be zero) as the patient moves a more or less straight line (i.e., the patient does not make a rotation around the vertical world axis). The uncertainty on that artificial constraint is set such that small fast rotations of the upper limb can happen (during one step), but longer term variations are constrained. Together with the third output, it also constraints drift around the vertical axis for the lower leg. Note that when the patient for one reason or another needs to move on a, e.g., circular line rather than on a straight line, the method can accommodate it, but the operator needs to explicate this in the method. The model output needs to correspond to the principal direction of motion when it is changing.
$$\begin{array}{ccc} z_{4} & = & \mathrm{atan}2\left(2\left(q_{0}\cdot q_{1}+q_{2}\cdot q_{3}\right),1-2\left(q_{1}^{2}+q_{2}^{2}\right)\right),\\ & = & g_{4}\left(\underset{states}{\underset{︸}{q_{w}^{A},q_{w}^{B}}}\right),\\ & = & \mathrm{artificially}\;\mathrm{set}\;\mathrm{equal}\;\mathrm{to}\;0, \end{array}$$
by using
$$q_{w}^{A}=\left[\begin{array}{cccc} q_{0} & q_{1} & q_{2} & q_{3} \end{array}\right].$$

The output is the Euler angle around the vertical axis and calculated using atan2 to disambiguate the sign of the angle around zero.

### 2.6. Application of the Rauch Tung Striebel Filter: Recursive Smoothing Filter

A Rauch Tung Striebel (RTS) filter is implemented using the recursive formulation. The model is not linear; hence, a standard extended implementation of the RTS filter (with Jacobians as a linear approximation of the models) will be used, as an analogue of the Extended Kalman Filter. Compared to the standard Kalman Filter, which only propagates in a forward direction, an RTS has an additional loop which propagates the data in a reverse loop, which is possible here since no online usage is required. Its advantages compared to a standard Kalman Filter are elaborated further in the text.
Rauch Tung Striebel filter**Model and observation**:$$\begin{array}{ccc} X_{k+1} & = & f\left(X_{k},U_{k}\right)+\epsilon ,\;\;\;\mathrm{where}\;\;\;\epsilon \in \;N\left(0,Q\right)\\ z_{i,k+1} & = & g_{i}\left(X_{k},P\right)+\varepsilon_{i}\;\;\;\mathrm{where}\;\;\;\varepsilon_{i}\in \;N\left(0,R_{i}\right)\\ & & \mathrm{for}\;i=1\ldots \mathrm{Amount}\;\mathrm{of}\;\mathrm{measurements}\\ z_{k+1} & = & \left[\begin{array}{cccc} z_{1,k+1} & z_{2,k+1} & \cdots & z_{N,k+1} \end{array}\right]\\ g & = & \left[\begin{array}{cccc} g_{1} & g_{2} & \cdots & g_{N} \end{array}\right]\\ A_{k} & = & \;\mathrm{Jacobian}\;\mathrm{of}\;f\;\mathrm{with}\;\mathrm{respect}\;\mathrm{to}\;X_{k}\\ B_{k} & = & \;\mathrm{Jacobian}\;\mathrm{of}\;f\;\mathrm{with}\;\mathrm{respect}\;\mathrm{to}\;U_{k}\\ H_{k} & = & \;\mathrm{Jacobian}\;\mathrm{of}\;g\;\mathrm{with}\;\mathrm{respect}\;\mathrm{to}\;X_{k} \end{array}$$**Loop 1 - Forward filter**: for $k=1\to N_{data}$after proper initialization$$\begin{array}{ccc} X_{k|k-1} & = & f(X_{k-1|k-1},U)\\ P_{k|k-1} & = & A_{k}P_{k-1|k-1}A_{k}^{T}+Q_{k}\\ y_{err} & = & z_{k}-g\left(X_{k|k-1},P\right)\\ S_{k} & = & R_{k}+H_{k}P_{k|k-1}H_{k}^{T}\\ K_{k} & = & P_{k|k-1}H_{k}^{T}\mathrm{inv}\left(S_{k}\right)\\ x_{k|k} & = & x_{k|k-1}+K_{k}y_{err}\\ P_{k|k} & = & \left(I-K_{k}H_{k}\right)P_{k|k-1} \end{array}$$**Loop 2 - Reverse filter**: for $k=N_{data}\to 1$after proper initialization$$\begin{array}{ccc} C_{k} & = & P_{k|k}F_{k}^{T}\mathrm{inv}\left(P_{k+1|k}\right)\\ X_{k}^{S} & = & X_{k|k}+C_{k}\left(X_{k+1}^{S}-X_{k+1|k}\right)\\ P_{k}^{S} & = & P_{k|k}+C_{k}\left(P_{k+1|k+1}-P_{k+1|k}\right), \end{array}$$

where
$$X_{k}=\left[\begin{array}{c} q_{w}^{A}\\ q_{w}^{B} \end{array}\right],\;\;\;\mathrm{and}\;\;\;U=\left[\begin{array}{c} \omega^{A}\\ \omega^{B} \end{array}\right],$$
and the set of parameters
$$P=\left[\begin{array}{c} j_{1}^{A},j_{2}^{B},p_{1}^{A},p_{1}^{B} \end{array}\right].$$

Because the uncertainty is propagated using Jacobians which are numerically determined and include dependency on measurement signals (speed), for stability, the two filter gains (forward and reverse) can be diminished by 10%, improving on robustness and giving only slightly in on accuracy. Drift of motion of either limbs around the vertical axis cannot be detected with accelerometer based sensors. The large flexion motion is relative easy to obtain as an potential error on the attitude of the frame only results in a cosine type error on the main flexion motion. The smaller angles (vagus ad/abduction), however, are extremely sensitive on the correction attitude of the reference frame; if the frame is not oriented properly, the large flexion motion leaks as a sine error into the smaller motions. Misjudgment of the vertical angle due to drift prohibited earlier approaches. The RTS filter allows to propagate information forward as a Kalman filter does (indicated by (*) in Figure 2) but also backward (indicated by (**) in Figure 2). This means that, if the vertical alignment of the two frames are perfectly (certain) known when the leg is stretched (or has any other arbitrary angle), this information propagates and counter acts drift after this moment but also eliminates drift before this moment. Since the time between steps taken is relatively short, the drift around the vertical rotation axis is limited. The overall results are more certain (about twice smaller band than a traditional Kalman filter), and the phase distortion when one of the sensors is providing uncertain information is less. This is illustrated in Figure 2.

The filter is run a first time yielding flexion angles while the covariance matrices linked to the vertical rotation are kept constant at a relative high value (uncertain). The first pass provides an accurate enough estimate of the flexion angle. A second pass is then taken, whereby the covariance matrices are made dependent on the (first pass) flexion angle. The result of the second pass is accurate for all three angles.

### 2.7. Post Processing

Post processing uses straightforward quaternion algebra. Knowing the quaternions of the upper and lower leg describing their rotation in a world reference frame, we can now have a look at the relative motion of the two members with respect to each other.
$$q_{A}^{B}=\mathrm{inv}\left(q_{w}^{A}\right)⊗q_{w}^{B}.$$

When the sensors are zeroed after installing on the body of the patient, we need to rotate first the results given in sensor coordinate system into the body (upper and lower leg) coordinate system. The attitude of the body with respect to the sensor frame can be determined graphically using a picture taken from the patient (with reference color markings drawn on the hip, knee, and ankle joint) upfront and sideways. There are no quality requirement for the camera (can be a smartphone), except that care must be taken that the camera is vertical when taking a picture.

## 3. Results

### 3.1. Human Analogue Test

In order to validate the proposed method, a first test is performed where known angle movements are estimated from measured IMU data.

Therefore, as a human analogue a robotic manipulator (ABB IRB 120) was used to simulate a large flexion/extension angle, and small varus/valgus and internal/external rotation angles. The robot is seen as a human leg which is turned upside down. The IMU’s are strapped to the upper leg analogue - IMU2 on link 2 - and lower leg - IMU3 on link 4. This is illustrated in Figure 3. The analogue of the upper leg was kept fixed while the analogue of the lower leg was moved. The large flexion/extension angle is realized by rotation of robot axis 3 while axis 4 is used to disturb this motion and generate varus/valgus and internal/external rotation angles. Motions of both axis are plotted in Figure 4 and visualized in Figure 3 (motion sequence: frame 1 - frame 2 - frame 3 - frame 1).

Both IMU data and robot joint axis data was logged. Figure 5 and Figure 6 show the results obtained from the RTS filter obtained from the IMU data, compared to the robot logged data. Figure 5 shows the quaternion values of the lower leg with respect to the world axis frame obtained from RTS filtered IMU data, in full lines, and robot data, in dashed lines. It can be seen that only a minor estimation error is made. Furthermore, leakage to the main motion is minimal. Figure 6 shows the roll, pitch and yaw angle representation of this quaternion. Again, only a minimal estimation error is made. A peak angle error of 1° is obtained which is more than satisfactory for clinical analysis of the motion. From this experiment, it can be concluded that all three angles can be estimated with sufficient precision by using the proposed approach.

### 3.2. Lunge Movement

Once the method has been benchmarked on a robotic manipulator (human analog) the method is validated on clinical data obtained from a lunge movement. Figure 7 shows the experimental setup. The IMU sensors are strapped to the patient above and below the knee joint and data was captured using the IMU compatible software.

Reflective markers (44) were placed according to the lower limb and trunk model (LLT) described by [11]. These markers were rigidly secured at the following anatomical locations: cervical vertebrae C7, sternum, xipoïd process, T8, both posterior superior iliac spines, both anterior superior iliac spines, both iliac crests, both acromions, both greater trochanter, the medial and lateral knee epicondyle, the medial and lateral malleolus, metatarsal head 1 and 5, and both heels. Tracking markers were firmly secured to the anterolateral aspect of both thighs (8) and shanks (8). Marker data were gathered at a sampling frequency of 120 Hz using an Optitrack flex 13 (six cameras) system (Naturalpoint, USA). All trials were visually inspected using Visual3D v5 software (C-Motion, Germantown, Canada). Kinematical data were filtered using a lowpass Butterworth filter with a cutoff frequency set at 6 Hz. Kinematics of the knee in the sagittal, frontal, and tranversal plane were computed using this set-up as a golden standard.

Figure 8 shows the results of a typical lunge movement, measured by both the IMU RTS approach and the camera system. Only the main flexion/extension angle is compared to the camera measurement since camera systems typically have issues in estimating varus/valgus and internal/external rotation angles. This has been discussed in [12,13]. The rotative motion does not generate a lot of motion that the camera can see (short lever arm between the dots and center of rotation); hence, the camera-based approach typically has difficulty identifying this motion. As can be seen from Figure 8, the main flexion/extension angle is identified properly.

The proposed approach has been benchmarked with the robot, proving good and reliable estimates; hence, it is assumed that good estimation of varus/valgus and internal/external rotation is obtained. The magnitude of the estimated varus/valgus and internal/external rotation is within the reasonable motion range and no outliers are observed. Although these motions are hard to interpret from a physical point of view, the authors do believe that the proposed approach provides good estimates of these secondary angles.

## 4. Conclusions

The method presented in this paper is relatively straightforward implementable, has a solid physical base such that the covariance matrices do not need to be estimated but have physical meaning, shows good results with respect to more elaborate state of the art camera-based techniques, and is scalable to other segments of the body (motion of the hip, arms, etc.). It enables both clinical and scientific evaluations of different patient groups or athletes on site with the use of small readily available sensors. It also opens up potential for analysis of the motion of top performance athletes. Further work includes an extension to a full body model and extension of the simple mechanical model to a dynamic model so that also working moments and resulting forces can be estimated.

## Figures and Tables

**Figure 1 sensors-20-01683-f001:**
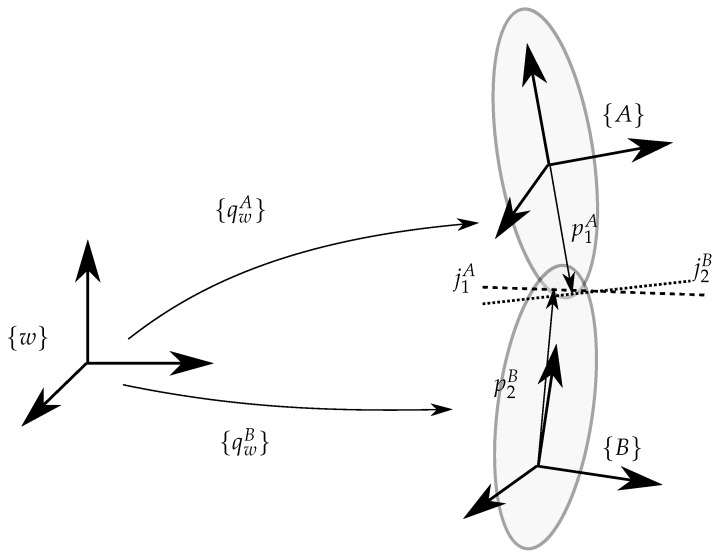
Schematic representation of the reference frames: world frame {w}, sensor frame {A}, sensor frame {B}, $q_{w}^{A}$ is the quaternion rotating the world reference frame into the reference frame {A}, $q_{w}^{B}$ is the quaternion rotating the world reference frame into the reference frame {B}.

**Figure 2 sensors-20-01683-f002:**
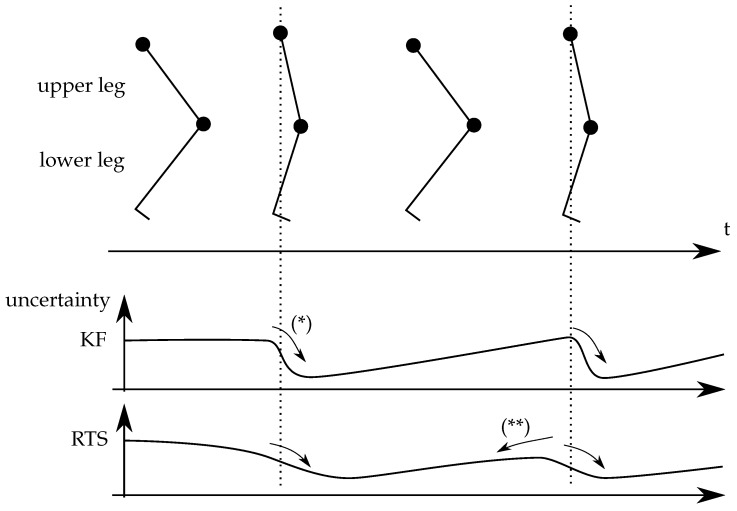
Schematic representation of the advantage of the Rauch Tung Striebel (RTS) compared to a standard KF. A RTS will smooth the uncertainty by using a ‘reference’ point (indicated by the vertical dashed line) and by traversing both ways (**), while a KF only traverse in one direction (*).

**Figure 3 sensors-20-01683-f003:**
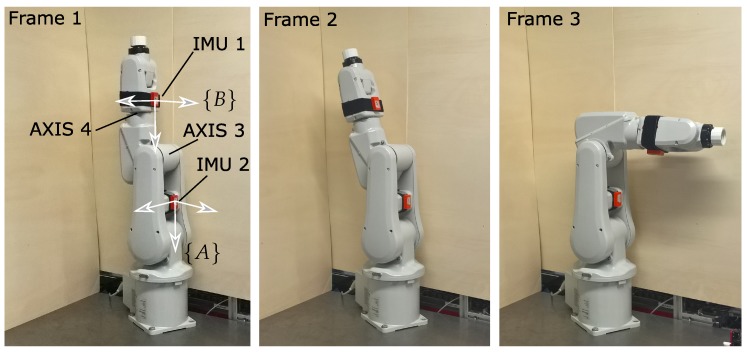
Illustration of the used human analogue setup: robotic serial manipulator ABB IRB-120 with inertial measurement units (IMU) sensors and reference frames. The motion sequence is Frame 1 - 2 - 3 - 1.

**Figure 4 sensors-20-01683-f004:**
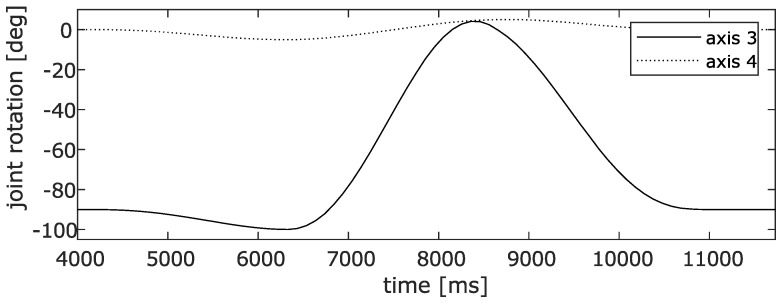
Angles Applied to the robot.

**Figure 5 sensors-20-01683-f005:**
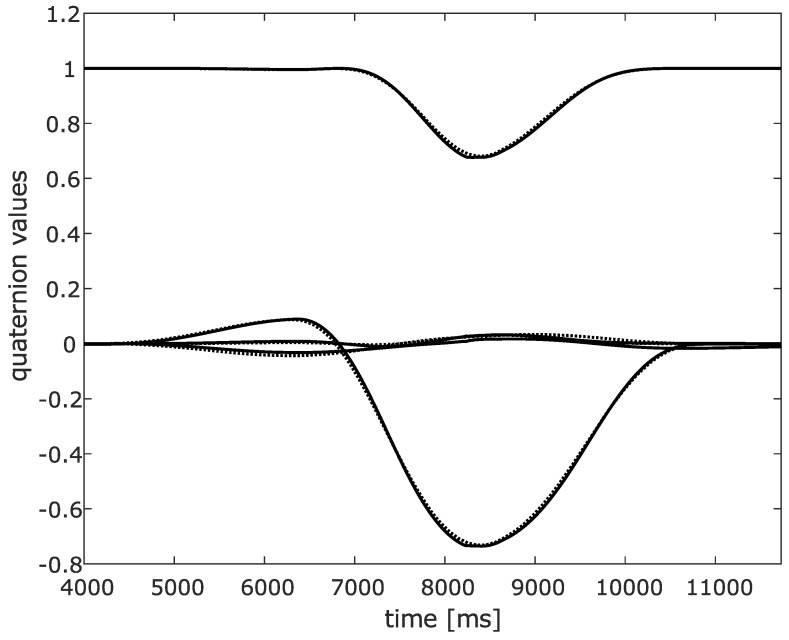
Quaternion values of the lower leg analogue of the RTS filtered IMU data (full lines) and robot logged values (dashed lines).

**Figure 6 sensors-20-01683-f006:**
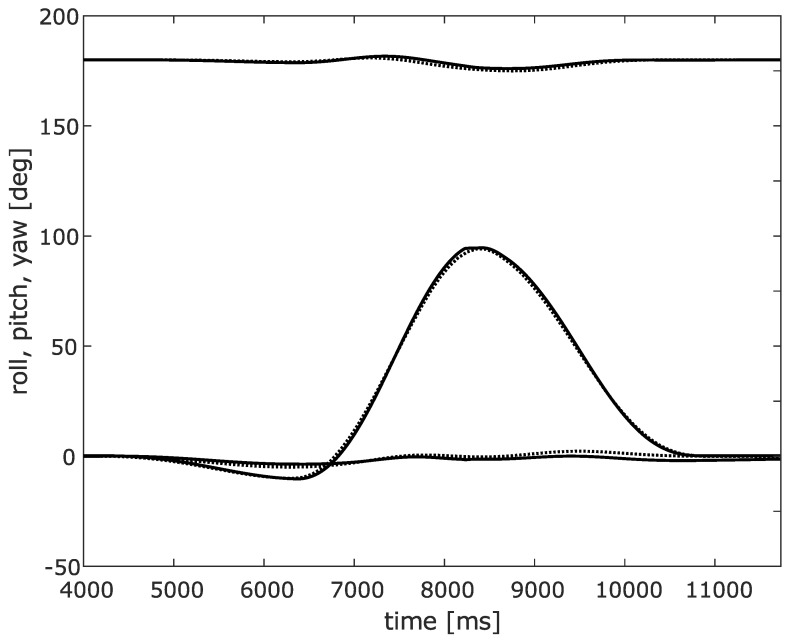
Roll, pitch yaw values of the lower leg analogue of the RTS filtered IMU data (full lines) and robot logged values (dashed lines).

**Figure 7 sensors-20-01683-f007:**
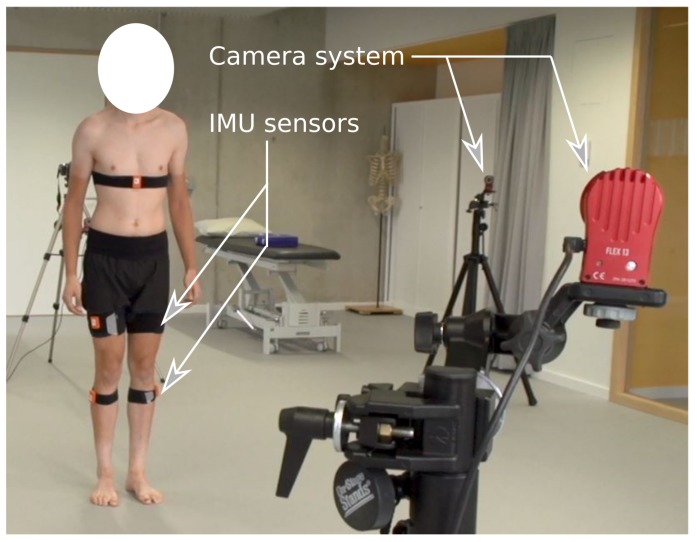
Experimental setup with visual markers for the camera system for comparison and IMU sensors for motion estimation with the proposed algorithm.

**Figure 8 sensors-20-01683-f008:**
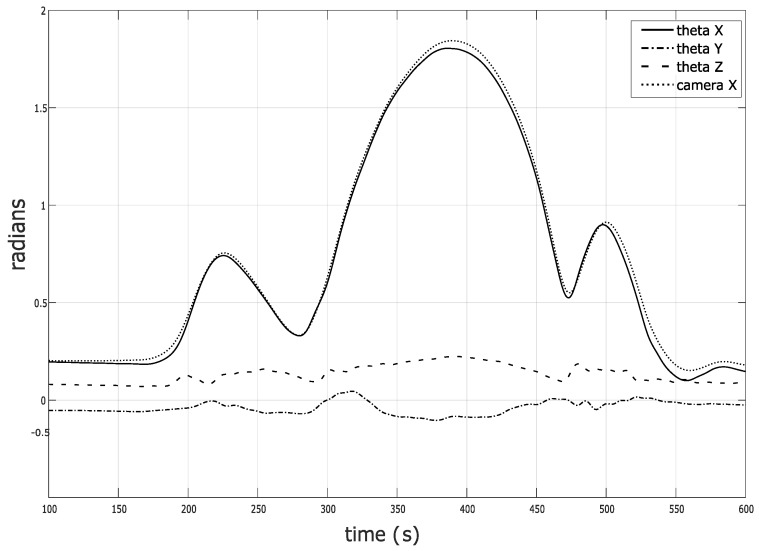
Estimated angles from a lunge movement. The estimated *X* (flexion/extension), *Y* (varus/valgus) and *Z* (internal/external rotation) Euler angles (thick, full, dashed, and dotted lines, respectively) are compared with the estimated angles obtained from the camera system (thin full lines).

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
