# Peer review of "A Novel Method to Estimate the Full Knee Joint Kinematics Using Low Cost IMU Sensors for Easy to Implement Low Cost Diagnostics"

_sensors, 2020, doi:10.3390/s20061683_

Round 1
Reviewer 1 Report
General comments:
I think the approaches presented will work well for sensor alignment and mitigating sensor drift. The use of smoothers should be pursued more for IMU-based motion capture. While I appreciate the concept, the paper appears to be hastly written with short introduction that does not cover relevant literature. Equations for Kalman filter implementation and joint alignment were not written in a manner to be directly replicable. The timeframe that the system were tested appeared to be very short,and perhaps short enough that dead-reckoned gyroscope measurements would give the same error.
Summary: This paper presents a method of estimating knee kinematics using IMUs.
Introduction: It is on the short side. There are many studies that have looked at
using IMUs for measuring lower extremity joint kinematics, but very few are mentioned. Several (including the ones that you mentioned)
proposed automatic segment alignment methods.
General Comments:
Line 125: How are you weighting the measurements?
What solvers are you using? How is the orientation determined after calculating the two primary axes?
Design of the Kalman filter:
How is the filter implemented? Extended Kalman Filter, multiplicative?
Is gyroscope measurements used in time model?
Is accelerometer measurements used in both the z1 and z2 constraints?
Is gyroscope and accelerometer bias considered?
Line 146: What is meant by the authors? You and your group?
Eq before line 155: is z1 suppose to match up with raw accelerometer measurements?
Line 211: What is (*) and (**)
What IMUs are used? Sampling rate? Sampling duration? Sensor calibration?
Resuls: Figure 5 and 6: figure does not comtain legends.
Results are not shown for for filtering approach
Figure 8: is the timeframe really 0.6 seconds? Seems really short.
Reviewer 2 Report
The paper describes a method for estimating full knee kinematics with low cost sensors and easy calibration procedures. The paper is potentially interesting, but it is really not well organised and written.
The introduction is not informative, there is no critical literature review indicating the strong points of the research and what is new and why; The description of the calibration is very difficult to follow There is no discussion on the limitation of the proposed method (in comparison with literature) How many subjects for the human test? Why just lunge and no walking activity? The results are not analysed properly. Need of more subjects and statistical analysisAuthor Response
Please see the attachment

Reviewer 3 Report
Dear Authors,
please amend the manuscript with the following changes:
-The literature review is exceptionally small, both in introduction and discussion section, please expand it
-please attach schematic diagram o the measurement system, state the equipment used, its parameters (response time etc.) and its uncerainty
-please try to establish expanded uncertainty of the performed measurements
-please comment on the inevitable time drift influence of the emu sensors
-please comment on the rigidity of the single-band mechanical connection of the sensor to the leg of the patient
sincerely,
Reviewer
Round 2
Reviewer 2 Report
The authors improved the paper consistently.